# HDX-MS reveals structural determinants for RORγ hyperactivation by synthetic agonists

Timothy S Strutzenberg, Ruben D Garcia-Ordonez, Scott J Novick, HaJeung Park, Mi Ra Chang, Christelle Doebellin, Yuanjun He, Rémi Patouret, Theodore M Kamenecka, Patrick R Griffin*

Department of Molecular Medicine, The Scripps Research Institute, Jupiter, United States

**Abstract** Members of the nuclear receptor (NR) superfamily regulate both physiological and pathophysiological processes ranging from development and metabolism to inflammation and cancer. Synthetic small molecules targeting NRs are often deployed as therapeutics to correct aberrant NR signaling or as chemical probes to explore the role of the receptor in physiology. Nearly half of NRs do not have specific cognate ligands (termed orphan NRs) and it's unclear if they possess ligand dependent activities. Here we demonstrate that ligand-dependent action of the orphan RORγ can be defined by selectively disrupting putative endogenous—but not synthetic—ligand binding. Furthermore, the characterization of a library of RORγ modulators reveals that structural dynamics of the receptor assessed by HDX-MS correlate with activity in biochemical and cell-based assays. These findings, corroborated with X-ray co-crystallography and site-directed mutagenesis, collectively reveal the structural determinants of RORγ activation, which is critical for designing RORγ agonists for cancer immunotherapy.
DOI: https://doi.org/10.7554/eLife.47172.001

*For correspondence: pgriffin@scripps.edu

Competing interests: The authors declare that no competing interests exist.

## Introduction

RAR-Related Orphan Receptor C (RORγ, gene name *RORC*) is an orphan nuclear receptor (NR) that is widely expressed and is involved in regulation of various metabolic processes (*Bookout et al., 2006*; *Takeda et al., 2012*). RORγ is also a key player in the immune system where the lymphocyte specific isoform RORγt is a so-called 'master regulator' of the IL-17 producing T helper (Th17) cell subset (*Ivanov et al., 2006*). Human patients with nonsense mutations are susceptible to candida infection of the lung (*Okada et al., 2015*), suggesting that the evolutionary pressure of RORγ weighs heavily on immune function. The development of ligands that repress RORγt action have been extensively pursued to treat Th17-mediated autoimmune disorders such as multiple sclerosis and psoriasis (*Bartlett and Million, 2015*; *Jones et al., 2012*). More recently, RORγ has also been found to be the driver of the androgen receptor in metastatic castration resistant prostate cancer where the selective RORγ antagonist SR2211 elicits a potent cytostatic effect (*Wang et al., 2016*). Conversely, ligands that activate RORγt have been implicated in improving Th17-mediated protective antitumor immunity (*Bartlett and Million, 2015*; *Jones et al., 2012*) but comparatively, their development has lagged.

While activating ligands of RORγt are present during T cell maturation (*Hu et al., 2015*), synthetic RORγ agonists have been found to further activate the receptor (hyperactivation) (*Chang et al., 2016*; *Marcotte et al., 2016*). Overexpression of RORγt is sufficient to induce IL-17 expression in Naïve CD4[+] T cells[3] although it is unclear if this activity is ligand dependent, or if it can be increased further with synthetic agonists. Interestingly, our lab and others have demonstrated that RORγ

agonists not only increase expression of IL-17 expression, but also decrease levels of programmed cell death one receptor (PD-1) (*Chang et al., 2016*; *Hu et al., 2016*). T cell surface PD-1 interaction with PD-L1 presented on tumor cell surfaces deactivates cytotoxic T cell responses and promotes regulatory T cell phenotypes in CD4[+] subsets. Therapies targeting PD-1 have been very successful in improving immune surveillance of tumors in the context of non-small cell lung cancer, metastatic melanoma, and renal cell carcinoma (*Swaika et al., 2015*). Indeed, chimeric antigen receptor expressing T cells treated with RORγ agonists potentiate tumor clearance activity in mouse models (*Hu et al., 2016*). These findings further implicate RORγ agonism as a potential therapeutic strategy not only to enhance protective anti-tumor immunity by increasing IL-17 expression but also attenuating immunosuppressive action through PD-1 stimulation with one compound in clinical development for cancer therapy (*Hu et al., 2016*; *Wilkins et al., 2017*). Recently, we developed a series of N-aryl-sulfonyl indoline orthosteric RORγ agonists that were optimized for potency and selectivity (*Doebelin et al., 2016*).

Structural and functional analysis has revealed that the RORγ ligand binding domain (RORγLBD) is comprised of 12 α-helices (H1-H12) and a β-sheet region (BSR) that responds to cholesterol biosynthetic pathway intermediates, hydroxysterols and sterol sulfates (*Hu et al., 2015*; *Jin et al., 2010*; *Santori et al., 2015*). While the intrinsic activity of the receptor has been presumed to be ligand-dependent, recent studies have called to question the basis of the receptor's activity in situ. These studies include the crystal structure and NMR solution of 'apo' RORγ (*Li et al., 2017*), and recent molecular dynamic simulations (*Sun et al., 2018*; *Yukawa et al., 2019*). Collectively, these studies have identified that the *gauche* rotomeric state of W317 stabilizes H11, H11' and H12 through extensive hydrophobic interaction networks formed with F486, F506, Y502, and H479 as the key drivers of RORγ hyperactivation by stabilizing the required Y502-H479 hydrogen bond. Herein, we explore the ligand-dependent activities of the RORγLBD using site-directed mutagenesis. We further explore the structural basis for ligand-mediated hyperactivation of the RORγLBD using a comprehensive structural characterization by hydrogen-deuterium exchange coupled with mass spectrometry (HDX-MS) and RORγLBD:ligand co-crystal structures. The results presented here indicate that the RORγLBD requires a ligand to stabilize the active conformer and that hyperactivating ligands enhance coactivator affinities by allosterically driving electrostatic intramolecular interactions between H12 and H4. Combined these observations are explained by a revised model of RORγ activation which is used to more accurately describe RORγ pharmacology.

## Results

### HDX-MS characterization of RORγ

To shed light on ligand-dependent activation of RORγ, we sought to build upon previous structural and computational studies by characterizing solution-phase solvent exchange kinetics of recombinant RORγLBD using differential HDX-MS. In-line pepsin digestion of RORγLBD and LC-MS analysis reliably yields 54 peptides that cover 98% of the primary sequence, *Figure 1—figure supplement 1A*. During a 10 s to 4 hr time course, the majority of the sequence exhibits intermediate to slow kinetics, or moderate protection to exchange (*Figure 2C* and *Figure 1—figure supplement 1D–1E*), while H10-H12 exhibits fast kinetics, or little protection to solvent exchange, *Figure 2D* and *Figure 1—figure supplement 1C*. In the presence of endogenous ligands or partial agonists, such as 25-hydroxycholesterol (25OHC) or SR19265, several regions of RORγ became protected to solvent exchange with the most dramatic being the BSR and H12 (*Figure 1F and G*). Other regions that constituted the orthosteric binding site, but also distal regions including H1 and H9, exhibited significant protection to exchange. HDX-MS analysis of the selective RORγ inverse agonist SR2211 revealed a distinctive profile. Specifically, RORγLBD bound to SR2211 exhibited protection to amides surrounding the orthosteric binding site, but SR2211 offered no protection to H12 suggesting that the activation function surface remains disordered. The co-crystal structure solution of the RORγLBD bound to SR2211 reveals a deactivation where the carbinol moiety pushes W317 into an alternative conformation that prevent H11' and H12 from nucleating. These findings are consistent with previously observed pharmacology (*Kumar et al., 2012*) and show that HDX-MS measurements can distinguish agonists from inverse agonists. To examine responses to coactivator peptides, Differential HDX-MS was performed using the binary RORγLBD:SR19265 complex treated with a synthetic peptide

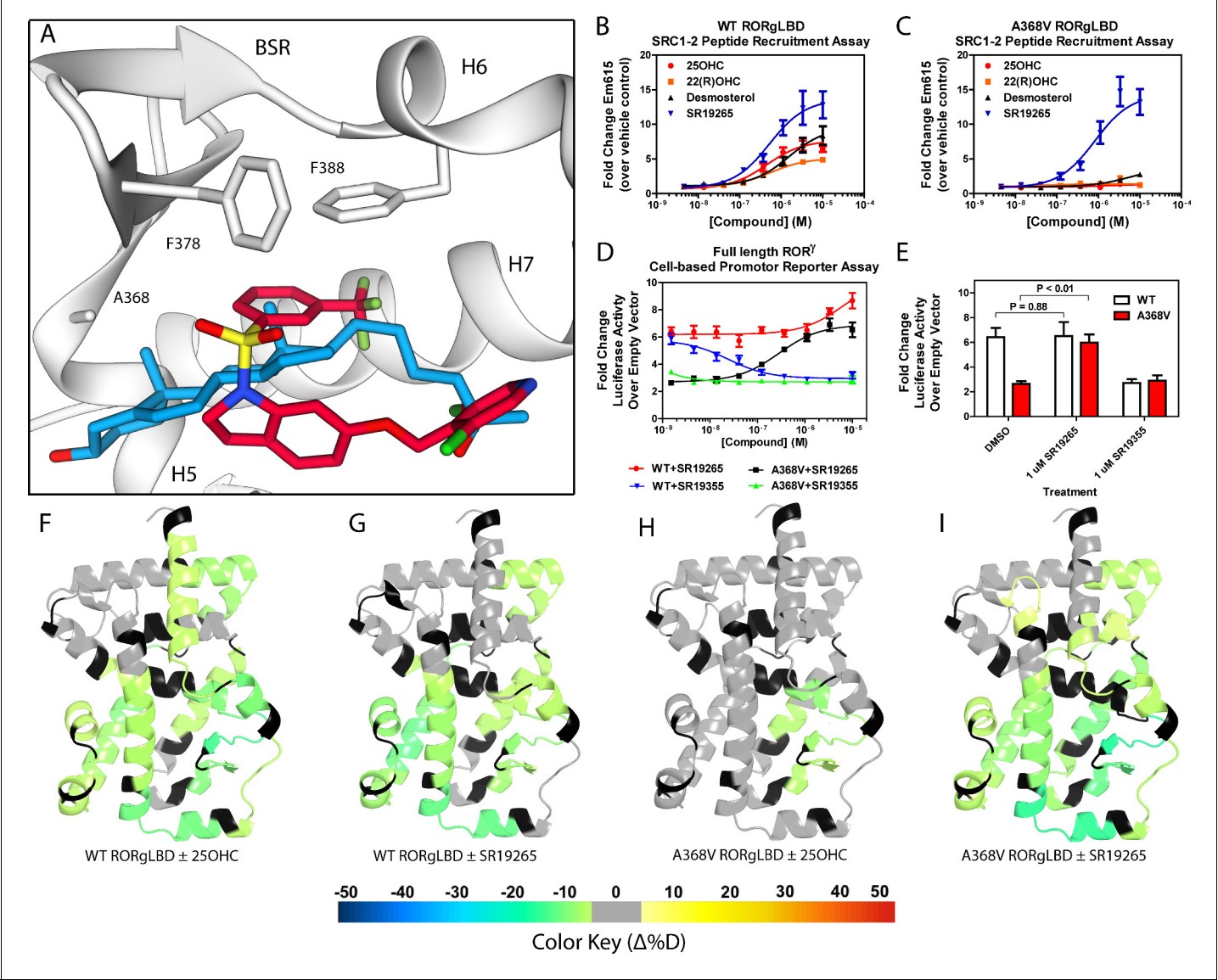

**Figure 1.** Ligand binding is required to activate RORγ in vitro. (**A**) The binding poses of SR19265 (red) and 25-hydroxycholesterol (blue, 25OHC) are compared. Both compounds are shown in the context of co-crystal structure solutions with RORγ (PDB ID 3L0L) where helices 5, 6, and 7 (H5, H6, and H7) as well as the beta sheet region (BSR) are shown. SR19265 and 25OHC are shown in red and blue respectively. A368V was found to be a mutation that selectively disrupts endogenous ligand binding presumably through steric clashes. WT and A368V RORγLBD were tested in an AlphaScreen-based SRC1-2 coactivator peptide recruitment assay in panels **B** and (**C**). (**D**) HEK293T cells were transiently transfected with a vector encoding full length RORγ and a 5xRORE-luciferase reporter. The WT and A368V variant were tested for their baseline activity as well as their response to SR19265 and SR19355. (**E**) Summary of luciferase assay activity showing A368V is a loss of function mutation and that activity is recovered with SR19265. Activity from WT and A368V variant RORγ are shown as white and red bars, respectively. (**F-I**) Solvent exchange kinetics of the WT (panels **F** and **G**) and A368V (panels **H** and **I**) variant RORγ LBDs were assessed using differential HDX-MS to compare 25OHC (panels **F** and **H**) and SR19265 (panels **G** and **I**) treated protein to vehicle control treated protein. The change in percent deuterium uptake of 54 peptides were averaged and consolidated for each amino acid and overlaid onto the RORγ LBD in the active conformation (PDB ID 3L0L). Gray and black regions of the protein indicate no significant changes to exchange and no sequence coverage, respectively.

DOI: https://doi.org/10.7554/eLife.47172.002

The following figure supplement is available for figure 1:

**Figure supplement 1.** Overview of HDX-MS characterization of RORγ (related to *Figures 1–5*).
DOI: https://doi.org/10.7554/eLife.47172.003

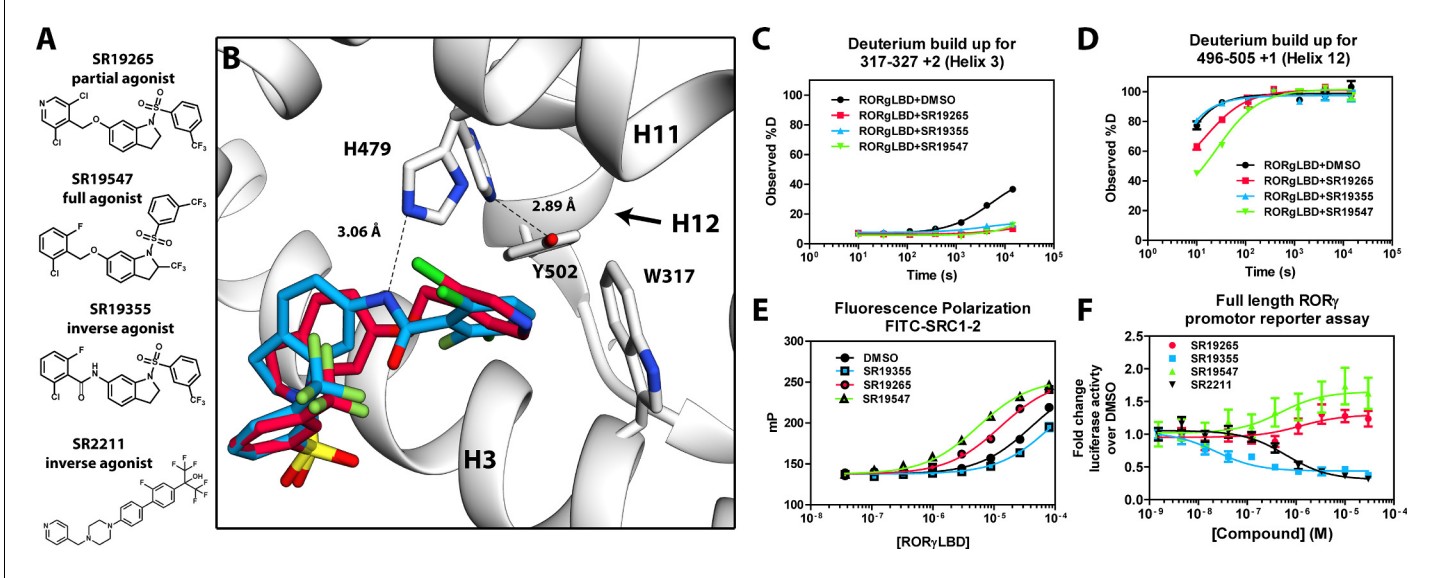

**Figure 2.** Mechanism of action for select N-arylsulfindoline RORγ modulators. (**A**) Representative structures of four modulators and their pharmacology. (**B**) Co-crystal structure solutions of SR19265 and SR19355 reveal mechanism of action. A peptide bond on SR19355 acts as a hydrogen bond donor for H479 and disrupts H479 and Y502 hydrogen bond. (**C, D**) Differential HDX-MS was employed to evaluate ligand dependent structural perturbations in solution. Representative deuterium build-up plots showing Helix 3 (**C**, H3) and Helix 12 (**D**, H12) dynamics. (**E**) fluorescence polarization results showing gradient of affinity for coactivator peptide depending on compound. (**F**) RORγ modulators were tested in a cell-based promotor reporter assay. Vectors encoding full length RORγ and a 5xRORE-Luciferase promotor reporter were transiently transfected into HEK293T cells. These cells were seeded into a 384 well plate and tested for dose-dependent responses to compounds.

DOI: https://doi.org/10.7554/eLife.47172.004

The following figure supplement is available for figure 2:

**Figure supplement 1.** Electron density maps are shown for compounds of interest (related to *Figures 1* and *2*).

DOI: https://doi.org/10.7554/eLife.47172.005

derived from steroid receptor coactivator 1 NR-box 2 (SRC1-2) or Tris-EDTA buffer (vehicle). The presence of SRC1-2 resulted in further protection to H4 and H10-12 consistent with published co-crystal structure solutions with coactivator peptides indicating ternary complex formation (*Jin et al., 2010*), *Figure 1—figure supplement 1B*. SRC1-2 was also able to protect these regions of the protein in the absence of SR19265 albeit to much lesser degree. Overall, the findings of these studies demonstrate the tractability between the structure of the protein and solution phase dynamics determined with HDX-MS. We also compared the dynamics of the RORγLBD-SRC2 fusion construct (RORγ265–507-GGG-SRC2-2), which was used to solve the 'apo' RORγ crystal structure, with our RORγLBD construct. This construct demonstrated similar kinetics of the ternary complex suggesting that the fusion construct artificially stabilizes the active conformer in solution, *Figure 1—figure supplement 1C and E*. Overall, this system exhibited behavior that was consistent with previous studies.

## RORγ is not constitutively active

The RORγ modulator SR19265 was characterized using an AlphaScreen-based coactivator peptide interaction assay and a cell-based promotor reporter assay. SR19265 was able to interact with a synthetic peptide derived from SRC1 NR-box 2 (SRC1-2) with similar potency and fold activation of the reported endogenous agonists 25OHC and desmosterol, *Figure 1B*. In cell-based assays, SR19265 was able to increase activity of a luciferase reporter above the baseline, *Figure 1D*. However, SR19265 exhibited a marked reduction in potency and the differences in activity were non-significantly different from the DMSO treated control group at a 1 μM dose compared to biochemical assays, *Figure 1E*. Examination of the co-crystal structure of RORγLBD bound to SR19265 revealed a distinct binding pose compared to a representative hydroxycholesterol 25OHC, *Figure 1A*. While 3-hydroxy and 3-sulfocholesterols are thought to be stabilized by hydrophobic contacts and hydrogen bonds with R367 (*Jin et al., 2010*; *Kallen et al., 2017*). SR19265 binding was located further away

from R367 and the interaction is driven strictly through hydrophobic contacts. Importantly, hydrophobic interactions observed by both compounds with BSR amino acids, F378 and F388, help explain dramatic protections to deuterium exchange to the β-sheet region (BSR) observed by HDX-MS. SR19265 makes hydrophobic contacts with W317 and H479 thereby stabilizing H10 and H12 through the W317-F486 π stacking and H479-Y502 hydrogen bond respectively, *Figure 2A*. To explain the observed differences in functional assays we tested both RORγ activation models (constitutive activity and activation by endogenous agonists) using site-directed mutagenesis. We hypothesized that a simple mutation to A368 could selectively disrupt hydroxycholesterol binding but not SR19265 binding. Using the SRC1-2 coactivator peptide interaction assay, we observed a marked reduction in hydroxycholesterol-mediated activation of A368V RORγLBD while SR19265 retained potency compared to the WT variant. In the cell-based promotor reporter assay, the A368V mutation is a loss of function where activity is recoverable with synthetic modulator, *Figure 1D*. The A368V mutant showed marked reduction in basal activity and exhibited a more potent response to SR19265. The presence of 1 µM SR19265 was able to recover activity to that of the WT receptor, *Figure 1E*. A possible explanation to the observed loss of function due to mutation of A368 to valine is that the valine residue disrupts the structural dynamics of the LBD. To test this hypothesis, we employed differential HDX-MS to examine changes in structural dynamics between A368V and WT RORγLBD, as well as the ligand dependent perturbations to SR19265 and 25OHC, *Figure 1F* through *Figure 1I*. Compared to WT RORγLBD, the A368V mutant showed faster solvent exchange in the BSR, H1, and H3. In the presence of 25OHC, the solvent exchange of these regions was reduced to levels similar to WT treated with vehicle and there was no stability introduced to H12, *Figure 1C*. On the other hand, A368V RORγLBD treated with SR19265, *Figure 1H*, showed similar protection to exchange as the WT RORγLBD treated with SR19265, *Figure 1F*, suggesting that the A368V variant liganded with SR19265 maintained similar structural dynamics to the WT receptor bound to SR19265.

## SR19547 and SR19355 are a full agonist and an antagonist respectively

To test this model, we began characterizing representatives of N-aryl sulfindoline, *Figure 2A*, class of RORγ modulators using co-crystallography, in vitro functional assays, and HDX-MS (*Doebelin et al., 2016*). In a fluorescence polarization-based coactivator peptide interaction assay, SR19265 and SR19547 treatment enhances the EC$_{50}$ of RORγLBD association with SRC1-2 to 12 µM and 1 µM respectively compared to 53 µM for the DMSO treated protein control, *Figure 2E*. The enhanced affinity for coactivator is consistent with the cell-based assays results where SR19547 is able to hyperactivate RORγ about 50–80% above the basal level, *Figure 2F*. Both SR19265 and SR19355 co-crystal structures were solved with 2.7 and 2.3 Å resolution respectively, *Figure 1A-Supplementary file 1* and *Figure 2A—figure supplement 1B-C*. In general, SR19265 and SR19355 share similar binding poses within the ligand binding pocket and make general hydrophobic contacts through the indoline and substituted benzoyl moieties with H3, H7 and H10 where protection to HDX is also observed, *Figure 2B and C*. Interestingly, the amide of SR19355 acts as a H-bond donor for H479, which disrupts the H479-Y502 H-bond leading to destabilization of H12. This finding is corroborated by HDX-MS where SR19265 treatment protects H12, whereas SR19355 treatment does not protect H12 to solvent exchange. As expected, SR19355 treatment further reduces coactivator peptide interaction, and like SR2211, SR19355 also robustly reduced RORγ activity in cells, *Figure 2F*. These results are consistent with previous analysis with SR2211 (*Kumar et al., 2010*). After obtaining co-crystal structures of partial agonist SR19265 and antagonist SR19355, we attempted and failed to solve co-crystal structures of full agonist SR19547 bound to the RORγLBD. Despite this, we were still able to study the effects of SR19547 on the dynamics of the RORγLBD where we see protection to H3, the BSR, with stronger protection to H12, *Figure 2C and D*, indicating that SR19547 strongly stabilizes the active conformation of RORγ. This finding supports a model where SR19547 increases coactivator affinity by presumably reducing entropic penalties of binding.

## Solution-phase dynamics correlate with biochemical and cell-based activity

We sought to further validate the model by extending structure-function analysis to the entire N-arylsulfonyl indoline class of RORγ modulators. To do this, we took advantage of quantitative high

throughput assays and HDX-MS screening. Interrogating the 38-compound dataset with statistical analysis enabled an unbiased look into the connection between structure and function. We used the AlphaScreen-based coactivator peptide interaction assay to sensitively monitor the changes in affinity for the SRC1-2 peptide. Compound activity in this assay was defined as emission at 615 nm (RLU) and measured at a single saturating concentration. The distribution of activity in this assay appeared normal, but left-tailed ranging from 45 to 20300 RLU (median of 10700), *Figure 3—figure supplement 1A-C*. Compound performance in the AlphaScreen-based peptide recruitment assay was found to correlate with affinity for coactivator peptide determined by fluorescence polarization, *Figures 2E* and *3B*, *Figure 3—figure supplement 1F*. Compound effects on the thermal stability of the RORγLBD was determined using differential scanning fluorimetry (DSF or thermal shift). In the thermal shift assay, the DMSO-treated receptor had a melt temperature of 45.0 °C and compounds shifted the melt temperature (ΔTm) by 4.2–13.2 °C (median of 7.8 °C) in a distribution that appeared normal, *Figure 3—figure supplement 1D*. To quickly measure ligand-dependent solution-phase dynamics, we developed a two timepoint HDX-MS screening approach to quickly assess compound effects on RORγ structural dynamics by only measuring perturbation at the 10 s and 4 hr timepoints. This strategy facilitated higher throughput and enabled us to measure perturbation signatures (a set of 54 peptides with perturbation values at two timepoints) for 38 compounds with five replicates. All compounds were found to generally engage the receptor in the orthosteric ligand binding pocket

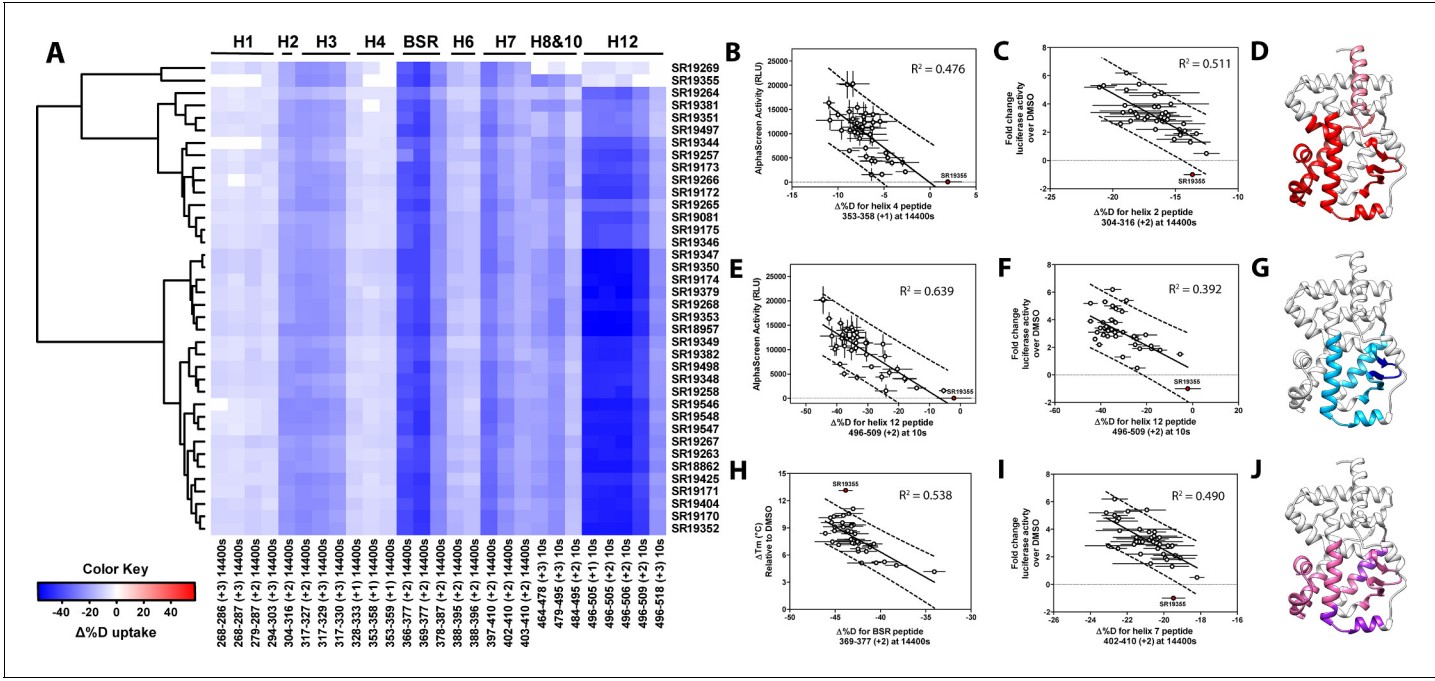

**Figure 3.** Differential HDX-MS and activity screening of 38 RORγ modulators reveals structural determinants for activation. (**A**) Δ%D (difference from DMSO treated control) values for 38 RORγ modulators are plotted as a heat map according to the color key at the bottom left. Each row represents a compound and each column represents a peptide indicated by the labels at the right and bottom respectively. The locations of each peptide in the crystal structure are labeled along the top. A hierarchical clustering algorithm based on Ward's method was performed to cluster compounds based on their exchange signatures and the corresponding dendrogram is shown on the left. (**B-J**) Activity correlation analysis results found that structural dynamics measurements correlate with functional activity. Regions of RORγ that correlate with coactivator peptide recruitment, thermal stability, and activation in cell-based assays are shown in panel D, G, and J, respectively. Lighter shades indicate that the slope of a linear regression was significantly (adjusted p value < 0.05) non-zero, while darker shades indicate an R (*Takeda et al., 2012*) was greater than 0.4. Representative coactivator peptide recruitment correlation plots for helix 12 and helix four are shown in B and E, respectively. The correlation between thermal stability and the BSR peptide are shown in panel (**H**). Representative correlation plots cell-based activity are shown for H2, H12 and H7 are shown in panels C, F, and I, respectively.

DOI: https://doi.org/10.7554/eLife.47172.006

The following figure supplement is available for figure 3:

**Figure supplement 1.** Summary and reproducibility of biochemical and two timepoint HDX-MS screening data (related to *Figures 3* and *4*).
DOI: https://doi.org/10.7554/eLife.47172.007

based on observed protection to H3-H7 and the BSR. The collection of ligands showed a wide range of deuterium exchange protection within H12. Compounds generally clustered into three groups; the first included compounds that did not stabilize H12 (likely antagonists or inverse agonists), the second group consisted of partial agonists that weakly stabilize H12, and the constituents of the third group strongly stabilized H12.

Before performing the covariation analysis, we first removed peptides with perturbation values that were not significantly different (Δ%D and false discovery rate <0.05%) from the DMSO treated control after applying a Benjamani-Hochberg multiple test correction. This filtered 108 peptide-time-points combinations to 29 peptides-timepoints that are shown in *Figure 3A*. Using this shorter list of peptides, we performed a covariation analysis to look for correlation between activity in functional assays. As expected, we observed that H12 stability at the 10 s timepoint correlated ($R^2 = 0.638$) with the ligand-dependent recruitment of SRC1-2, *Figure 3E*. We also observed that perturbation values of H4 at the 4 hr timepoint correlated ($R^2 = 0.476$) with coactivator recruitment activity, *Figure 3B*. Furthermore, protection to deuterium exchange in H2, H3, H7, the BSR, and H10 also correlated with coactivator peptide activity, *Figure 3D*. Linear regression between perturbation values of several H1 peptides and AlphaScreen activity were found to have correlation coefficients that were significantly different from zero (adjusted P value < 0.01) but were non-predictive ($R^2 < 0.5$). Overall, these measurements indicated some relationship between dynamics of these regions with activation of RORγ in vitro. In general, ligand dependent thermal stabilities and perturbation values showed weaker correlation. While perturbation values of peptides from four regions (H2, H3, the BSR, and H7), *Figure 3G*, were found to have with significantly nonzero correlation coefficients in a linear model, only protection to the BSR showed somewhat predictive correlation ($R^2 = 0.538$), *Figure 3H*. Importantly, SR19355 displayed much higher performance in this assay with no commensurate increase in protection at these regions. This highlights an important limitation to the approach where differences in binding mode (an additional hydrogen bond in this case) can identify compounds as outliers. The linear models describing thermal stability as the predicted value only described distributions that exclude SR19355. Next, we sought to correlate the biophysical measurements with activity of compounds in a cell-based promotor-reporter assay. The compounds were tested in a GAL4-RORγLBD co-transfection format using UAS-luciferase as a reporter gene. Cells were treated with 2 μM ursolic acid to repress the activity of RORγ prior to addition of test compounds, and activity of compounds were presented as fold change over DMSO-only control. The activity in such an assay for the compound collection has already been reported as fold change in luciferase activity compared to DMSO treated control (*Doebelin et al., 2016*). Compounds afforded similar patterns to the peptide coactivator recruitment assay, but correlation trends were somewhat different. Perturbation values of peptides originating in H2, H3, the BSR, H7, and H12 were found to have significantly non-zero correlation coefficients, *Figure 3J*. Interestingly, protection to H12 became a less significant predictor ($R^2 = 0.392$) of activity within cells, *Figure 3F*. Instead, activity in cell-based assays was best predicted by protection to H2 ($R^2 = 0.512$) and H7 ($R^2 = 0.496$), *Figure 3C and I*. Compounds that protected these regions to deuterium exchange typically had a higher fold change in reporter gene activity compared to cells treated only with DMSO.

To assess if regions of the protein have concerted motions, we performed an analysis of covariation to look for peptides within the LBD that have percent deuterium incorporation of equal magnitude. The correlation coefficient ($R^2$) is plotted as a heatmap in *Figure 4A—figure supplement 1*. Peptides with overlapping sequences typically correlated with each other. We found that ligand-dependent perturbation values clustered into three groups, *Figure 4B*. The first group correlated with H12 and consisted of H1, H4 and H10. While most compounds protected H12, SR19269 and SR19355 exhibited little protection to this region. This helped to determine whether the correlations were coincidental, or dependent on H12 stability. Since SR19355 and SR19269 were clear outliers in the H12-H1 correlation plot, *Figure 4D*, this correlation was likely coincidental where H1 happens to be more stable when H12 is stable. However, SR19355 and SR19269 also exhibit commensurate H4 protection showing that the observed H4 stability depends on H12 stability. Furthermore, a correlation coefficient of 0.779 suggests that a simple linear model explains most of the variance of the distribution, *Figure 4C*. The second cluster were correlated strongly ($R^2 > 0.7$) with the BSR and consisted of H3 and H7. SR19269 and SR19355 did not appear to be obvious outliers with the second group. This region is likely where compounds were directly engaging the protein, so it is unclear that the stabilities of the region depended on each other or were simply driven by varying degrees

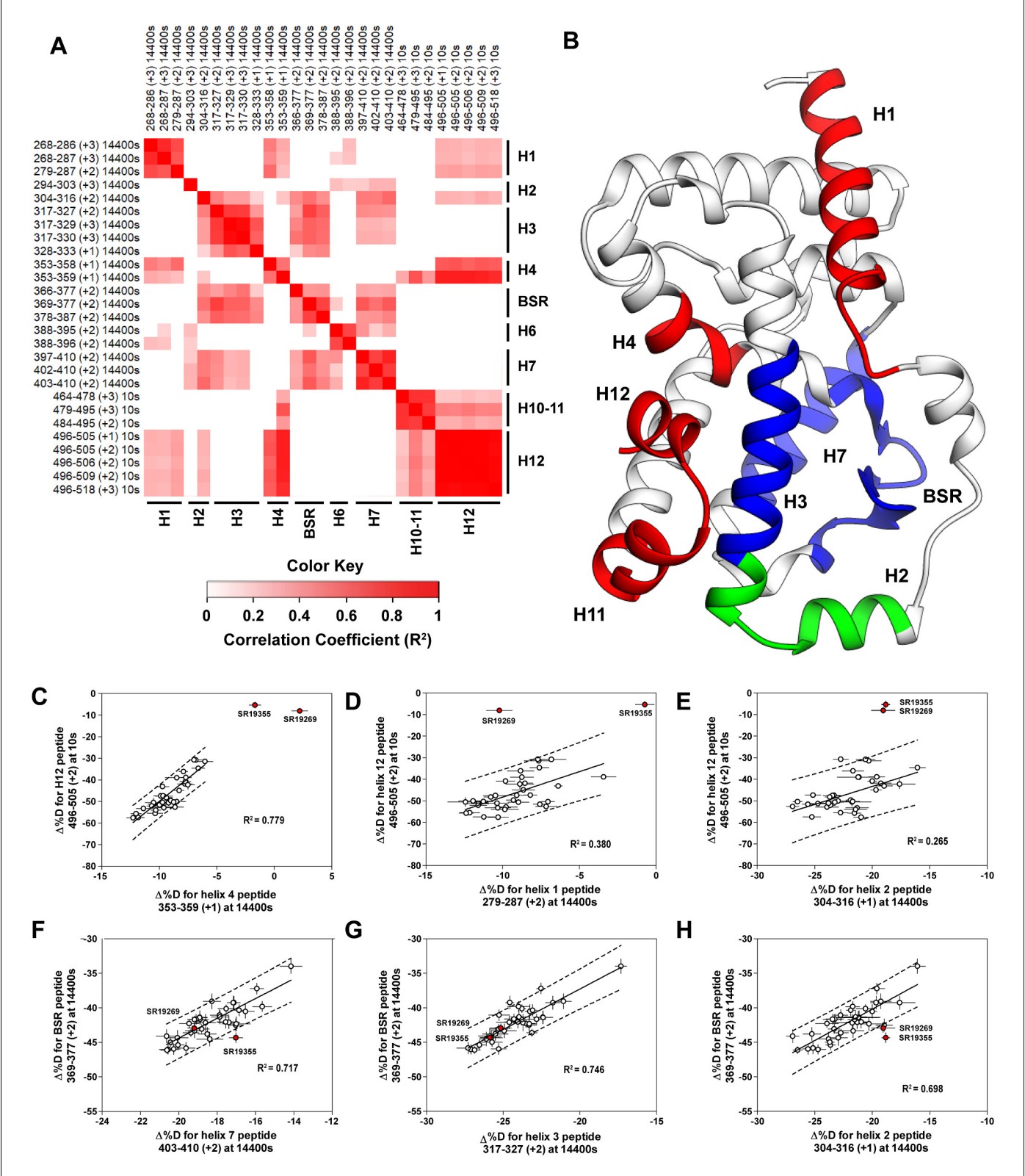

**Figure 4.** Covariation analysis of two timepoint HDX-MS screening reveals concerted ligand-dependent changes in protein dynamics. (A) Correlogram showing correlation coefficients ($R^2$) of Δ%D values between peptides. Peptides are labeled as start-end (charge) timepoint across the top and left sides of the panel while the location of the peptides are labeled along the bottom and right sides of the panel. Correlation coefficients that are significantly non-zero (adjusted P values < 0.01) slope are colored. (B) Perturbation values generally correlated in three groups. Group one correlated with H12 and

*Figure 4 continued on next page*

Figure 4 continued

consists of H1 and H4 and is colored red. Group two correlates with the BSR consists of H7 and H3 and is colored Blue. A peptide spanning residues 304–316 is shown in green. (C-H) Representative correlation plots showing distributions of compound perturbation values across peptides indicated by the x- and y-axis. Linear regression models are shown as solid black lines and the 95% confidence interval for prediction is shown as dashed lines. SR19355 and SR19269 (indicated in red) were often found as outliers for models involving helix 12 (panels C-E).

DOI: https://doi.org/10.7554/eLife.47172.008

The following figure supplement is available for figure 4:

**Figure supplement 1.** A weighted correlation analysis of the two timepoint compound screening dataset shows widespread correlation between peptides.

DOI: https://doi.org/10.7554/eLife.47172.009

of ligand binding. The third group consisted of a single peptide from H2, shown in green in *Figure 4B*, which correlated with both H12 and the BSR. Perturbation values of this peptide correlated strongly peptides from the BSR ($R^2$ = 0.698), *Figure 4H*. Given that SR19269 and SR19355 were clear outliers, the correlation with H12 appeared to be coincidental, *Figure 4E*.

## A network of electrostatic interactions is required to activate RORγ

Given the correlation with activation in vitro, we sought to further explore the relationship of H12 and H4 dynamics. To do this, we identified candidate intramolecular interactions formed by K354 and K503 by examining the structure. Both residues are hydrogen bond donors to backbone carbonyl as depicted in *Figure 5A*. Using site directed mutagenesis in combination with functional assays and HDX-MS, we characterized alanine and arginine mutants at both sites to study the role of the residues in receptor activation. Introducing either K503A or K354A variations to a GAL4-RORγLBD resulted in a loss of function in cell-based assays, *Figure 5B*. Mutation to K503R, but not K354R, was able to retain activity similar to WT levels, indicating general electrostatic interactions at K503 but not K354 are sufficient for activation. Treatment with synthetic agonist SR19547 was able to recover activity of the K503A construct, but not the K354A mutant suggesting the H12-H4 interaction is essential to activate RORγ in situ. These findings are reflected with in vitro coactivator interaction assays where K503A and K354A are less active than the WT control in the presence of 25OHC, shown in *Figure 5D*, respectively. Treatment with synthetic agonist SR19265 was able to recover some activity of the K503A construct and K354A at high concentrations suggesting the interaction between H12 and H4 is essential to activate RORγ in situ, while intramolecular K503 interaction is only required for activation by endogenous ligands. To further evaluate the structure-based model, we employed HDX-MS to examine the changes in solvent exchange due to loss of function mutations K354A and K503A, *Figure 5E–5I*. Comparing WT RORγLBD + SR19265 to K354A RORγLBD + SR19265 reveals no significant differences in deuterium exchange to peptides surrounding the orthosteric binding pocket. Specifically, the exchange signature of the BSR was identical across all three constructs indicating ligand binding, *Figure 5I*. Strikingly, protection to H10-12 was disrupted in varying degrees between the variant receptors with K354A being the most severe, *Figure 5E and H*. At the 10 s timepoint, H12 of K354A RORγLBD is completely deuterated, *Figure 5H*, much like apo WT RORγLBD. Furthermore, the K354A variant exhibits faster kinetics at H4 consistent with previous findings in the covariance study. Taken together, these observations suggest decoupling of ligand-binding from ligand mediated activation.

## Discussion

NR structure-activity relationship studies seek to evaluate modulator activity as a function of compound structure. While these studies reveal interesting trends in small molecule development, they fail to explain how the compounds modulate the activities of the receptor. Here we demonstrate that ligand-dependent conformational changes, assessed with HDX-MS, can empirically correlate with functional assay activity to reveal structural determinants for various biochemical activities. The N-arylsulfonyl indoline class of RORγ modulators exhibit disparate activities in biochemical and cell-based assays. The disparity between activation of RORγ in vitro and in cells was evident by the strength of the correlations for different regions of the protein. Ligand-dependent activation of the receptor in vitro and in cells both correlated with H12 stability. Interestingly, stability of H2 was the

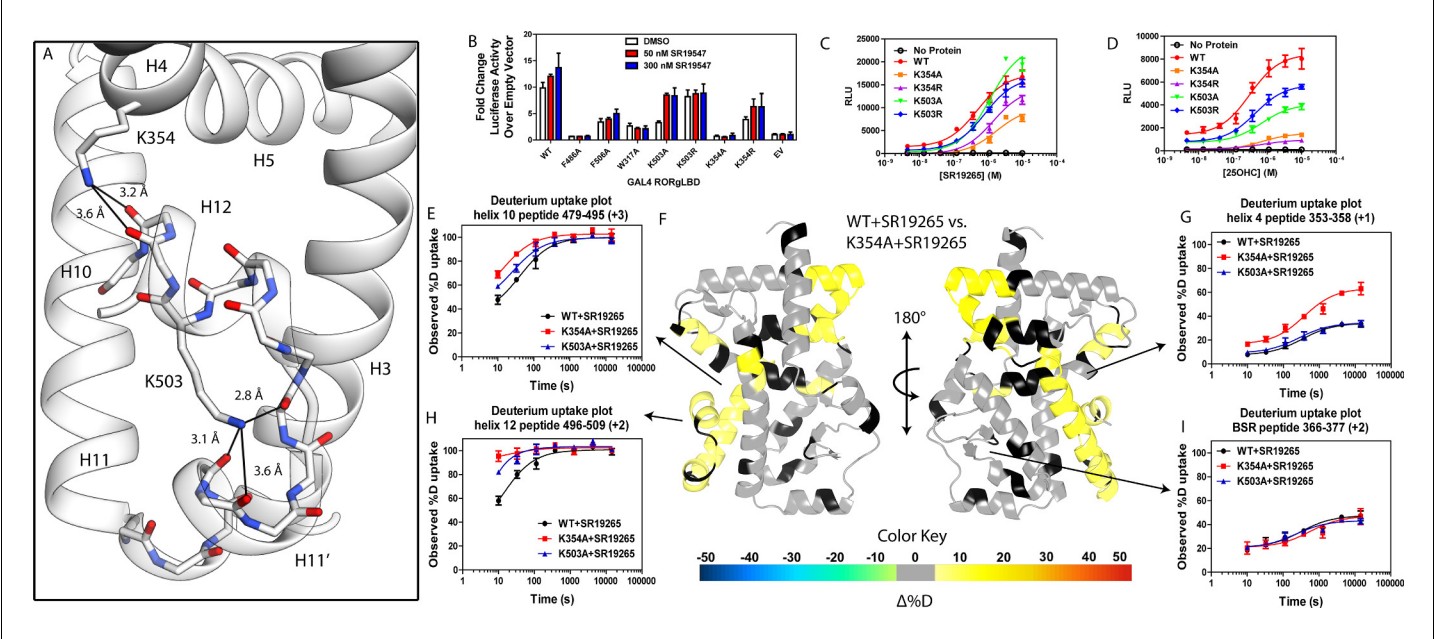

**Figure 5.** intramolecular interactions are necessary for activation by endogenous agonists. (A) RORγ in the active conformation (PDB ID 3L0L) reveals that K354 and K503 form several hydrogen bonds with backbone carbonyl shown in red. The functional consequences of these interactions in stabilizing the active conformation were assessed with site directed mutagenesis, promotor-reporter assays, (B), coactivator peptide recruitment assays (C and D) and HDX-MS (E-I). (B) RORγ variants were tested in a GAL4 chimera promoter reporter assay using a dual-glo assay format. SR19265 and 25OHC were tested for dose-dependent responses in AlphaScreen-based SRC1-2 coactivator peptide recruitment assay in panels C and D respectively. Solvent exchange kinetics of WT and mutant RORγLBDs bound to SR19265 were compared using HDX-MS in panels E-I. Deuterium uptake plots for helix 10, 4, 12 and the BSR are shown in panels E, G, H, and I, respectively. Results from differential HDX-MS comparing WT + SR19265 and K354A + SR19265 were consolidated and painted onto the structure of RORγ (PDB ID 3L0L) in panel G.

DOI: https://doi.org/10.7554/eLife.47172.010

best predictor of activity in cells. This is likely due to several biochemical activities required to hyper-activate the receptor *in cellulo*. As H2 stability generally correlated with H3, the BSR, H7 and H12, it is likely that stability of this region of the protein could serve as general indicator of global stability and activation potential. The results presented here suggest that the ligand-dependent conformational changes originate at the BSR and H3 interface. Not only did every modulator stabilize these regions, but the stability of H3 and the BSR correlated with ligand-dependent stability of the receptor. Interestingly, SRC2 fusion to H12 mimics ligand-dependent stability around the distal orthosteric binding site including the BSR despite being 5 nm away. In addition, the use of mutagenesis to selectively disrupt H10-12, but not the orthosteric binding site suggests that several intramolecular interactions are required to stabilize AF2.

To build upon RORγ agonism as a therapeutic strategy, we have better characterized the basis of the high basal activity of RORγ and the structural basis of RORγ activation. While RORγ is able to bind a coactivator peptide with excessive amounts of protein in vitro, this low affinity interaction is likely not biologically relevant. Experiments reported here show that the SRC2 fusion construct used to solve the 'apo' crystal structure artificially locks RORγ in the active conformation. To address the ligand-dependent activity of RORγ, we employed a 'bump-and-hole' strategy where A368V RORγLBD does not respond to endogenous ligand and has little to no intrinsic activity in promotor-reporter assays. An important limitation to this strategy is that the A368V mutation may disrupt both the binding and activation by endogenous ligands but also the structure of the 'apo' protein responsible for the intrinsic constitutive activity. The results of our mutational studies can only be used to make inferences and therefore cannot be taken as direct proof that the WT receptor requires ligand for activity. Regardless, both WT and A368V RORγ show similar deuterium exchange kinetics suggesting similar conformational ensemble, have little intrinsic activity in vitro, and are able to be activated by synthetic ligands that can bind the mutant construct. Taken together, these observations

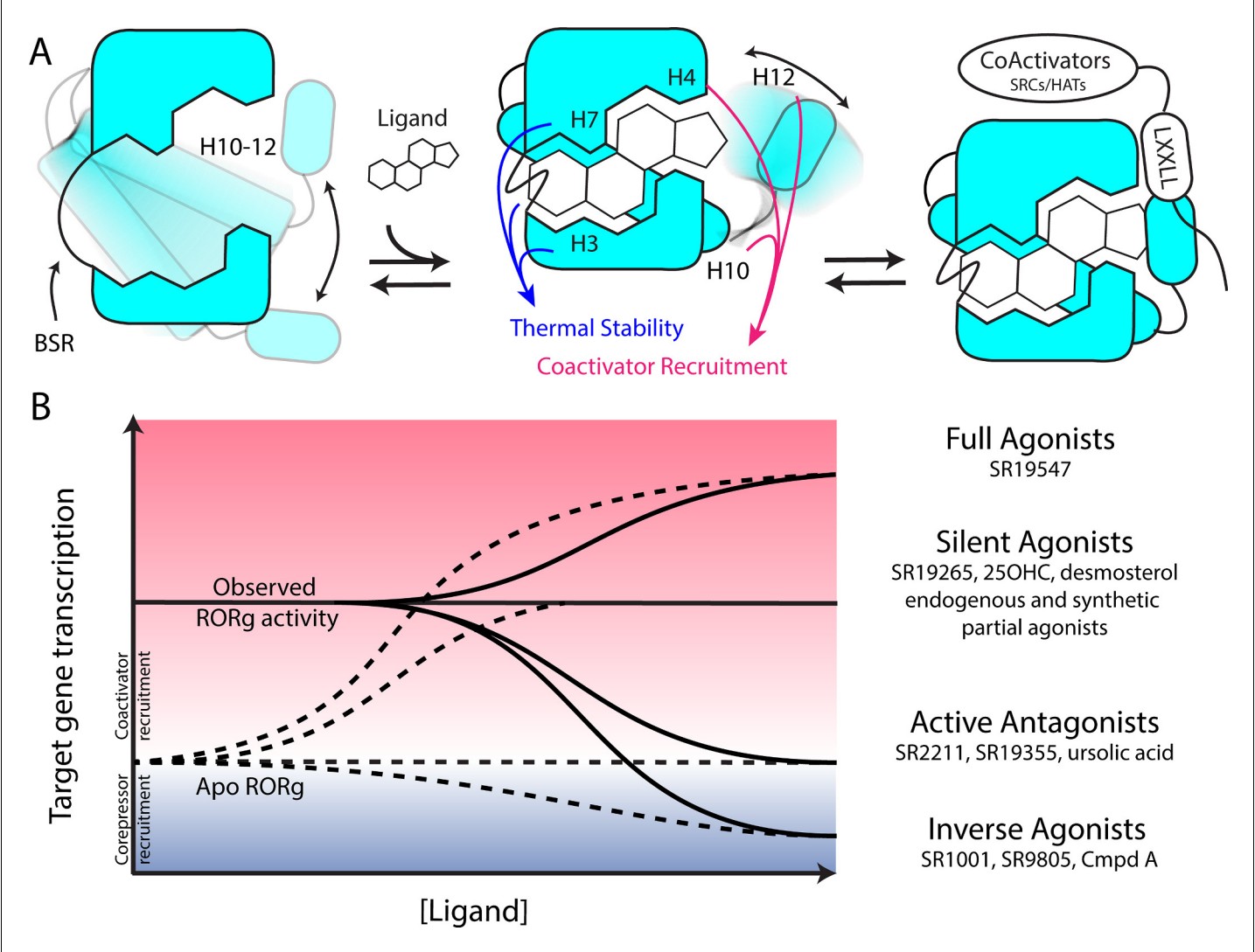

**Figure 6.** Model for ligand-dependent activation of RORγ. (**A**) Apo RORγ exhibits extensive conformational dynamics in the absence of compound. Ligand recognition through the BSR, H3 and H7 confer thermal stability and allows for H10-12 stability where H12 and H4 dynamics are the key structural determinants for coactivator affinity and receptor hyperactivation. (**B**) Based on mutagenesis studies, apo RORγ is presumed to be inactive. The observed high basal activity is likely due to activation of RORγ by endogenous agonists. Compounds that outcompete these endogenous ligands and activate RORγ to the observed levels should be considered silent agonists.

DOI: https://doi.org/10.7554/eLife.47172.011

are collectively consistent with the model where endogenous ligands are required for the observed activity in cells. A more accurate representation of apo RORγ is likely a dynamic ensemble where the active conformer likely does not exist for prolonged periods of time within the biological context.

Based on the two-step activation model, a more accurate depiction of unliganded RORγ is a dynamic ensemble of conformations where H12 makes a disordered-to-ordered transition upon ligand binding and is further stabilized with coactivator peptide binding. While this model of NR activation is commonly observed throughout the NR structural biology field (*Rastinejad et al., 2015*), we have elaborated on the model to elucidate the structural determinants for the gradient of observed activities exhibited by RORγ agonists, *Figure 6A*. According to this model, RORγ agonists must not only out compete endogenous ligands, but also stabilize a conformer that is more productive in driving association with coactivators than the endogenous agonists in order to hyperactivate the receptor. Modulators that activate RORγ to the same level should be considered silent agonists, *Figure 6B*.

# Materials and methods

### Key resources table

| Reagent type (species) or resource | Designation | Source or reference | Identifiers | Additional information |
|---|---|---|---|---|
| Cell line (HEK293T) | HEK293T | ATCC | Cat:CRL-11268 | |
| Transfected construct (Human) | pBIND-RORγ Hinge-LBD | Kumar et al., 2010 DOI: 10.1124/mol.109.060905 | Addgene_128091 | |
| Transfected construct (Human) | pGL4.35[luc2P/9XGAL4 UAS/Hygro] Vector | Promega | Cat:E1370 | |
| Transfected construct (Human) | pSPORTV6-hRORγ | Kumar et al., 2012 DOI: 10.1021/cb200496y | Addgene_128093 | |
| Transfected construct (Human) | pGLA4.1-5xRORE-Luciferase | Kumar et al., 2012 DOI: 10.1021/cb200496y | Addgene_128094 | |
| Commercial assay or kit | Britelite Luciferase assay reagent | Perkin Elmer | Cat:6066766 | |
| Commercial assay or kit | Dual-Glo Luciferase assay reagents | Promega | Cat:E2920 | |
| Recombinant DNA reagent | pESUMO-RORγLBD | Doebelin et al., 2016 DOI: 10.1002/cmdc.201600491 | Addgene_128090 | |
| Recombinant DNA reagent | pESUMO-RORγ LBD-SRC2 | This Paper | Addgene_128089 | |
| Commercial assay or kit | Q5 mutagenesis kit | New England Biolabs | Cat:E0554S | |
| Sequence-based reagent | Q5_hRORC_F378A_Sense | Sigma | N/A | CACGGTCTTTgcg GAAGGCAAATAC |
| Sequence-based reagent | Q5_hRORC_F378A_Antisense | Sigma | N/A | CGGTTGTCAG CATTGTAG |
| Sequence-based reagent | Q5_hRORC_F388A_Sense | Sigma | N/A | CATGGAGCTGgcg CGAGCCTTGG |
| Sequence-based reagent | Q5_hRORC_F388A_Antisense | Sigma | N/A | CCACCGTATT TGCCTTCA |
| Sequence-based reagent | Q5_hRORC_F486A_Sense | Sigma | N/A | GCTGCAGATCgcg CAGCACCTCC |
| Sequence-based reagent | Q5_hRORC_F486A_Antisense | Sigma | N/A | CTTTCCACATG CTGGCTAC |
| Sequence-based reagent | Q5_hRORC_K503A_Sense | Sigma | N/A | TCCACTCTACgcg GAGCTCTTCAGCACTG |
| Sequence-based reagent | Q5_hRORC_K503A_Antisense | Sigma | N/A | GGGAAAGCG GCTTGGACC |
| Sequence-based reagent | Q5_hRORC_K503R_Sense | Sigma | N/A | TCCACTCTACcgt GAGCTCTTCAGC ACTGAAAC |
| Sequence-based reagent | Q5_hRORC_K503R_Antisense | Sigma | N/A | GGGAAAGC GGCTTGGACC |
| Sequence-based reagent | Q5_hRORC_F506A_Sense | Sigma | N/A | CAAGGAGCTCgcg AGCACTGAAACC |
| Sequence-based reagent | Q5_hRORC_F506A_Antisense | Sigma | N/A | TAGAGTGGAG GGAAAGCG |

*Continued on next page*

*Continued*

| Reagent type (species) or resource | Designation | Source or reference | Identifiers | Additional information |
|---|---|---|---|---|
| Sequence-based reagent | Q5_hRORC_Y502F_Sense | Sigma | N/A | caagGAGCTCT TCAGCACTGAAACC |
| Sequence-based reagent | Q5_hRORC_Y502F_Antisense | Sigma | N/A | aagagTGGAGG GAAAGCGGCTTG |
| Sequence-based reagent | Q5_hRORC_A368V_Sense | Sigma | N/A | GATGTGCCGGg tgTACAATGCTGAC |
| Sequence-based reagent | Q5_hRORC_A368V_Antisense | Sigma | N/A | CTAACCAGC ACCACTTCC |
| Sequence-based reagent | Q5_hRORC_K354A_Sense | Sigma | N/A | TGTGCTTCTCgcc GCAGGAGCAATG |
| Sequence-based reagent | Q5_hRORC_K354A_Antisense | Sigma | N/A | ATCTGGTCATT CTGGCAG |
| Sequence-based reagent | Q5_hRORC_K354R_Sense | Sigma | N/A | TGTGCTTCTCcgg GCAGGAGCAATG |
| Sequence-based reagent | Q5_hRORC_K354R_Antisense | Sigma | N/A | ATCTGGTCAT TCTGGCAG |
| Peptide, recombinant protein | Biotin-Linker-SRC1-2 | Lifetein | N/A | Biotin-Ahx-SPSSH SSLTERHKILHR LLQEGSP |
| Peptide, recombinant protein | HisSUMO-hRORgLBD (265-518) | *Doebelin et al., 2016* DOI: 10.1002/ cmdc.201600491 | N/A | |
| Peptide, recombinant protein | RORgLBD (265-507) | This Paper | N/A | For co-crystallography with SR2211 |
| Peptide, recombinant protein | RORgLBD (265-507)-linker-SRC2-2 | This Paper | N/A | For co-crystallography with SR19265 and SR19355 |
| Commercial assay or kit | AlphaLISA Anti-HIS-Acceptor beads | Perkin Elmer | Cat:AL128C | |
| Commercial assay or kit | AlphaScreen Streptavidin-Donor beads | Perkin Elmer | Cat:6760002S | |
| Chemical compound, drug | Fc1cccc(Cl)c1O Cc2ccc3ccn(c3c2)S(=O) (=O)c4cccc(c4)C(F)(F)F | *Doebelin et al., 2016* DOI: 10.1002/ cmdc.201600491 | ID:SR18862 | |
| Chemical compound, drug | Fc1cccc(Cl)c1OCc 2ccc3CCN(c3c2)S(=O) (=O)c4cccc(c4)C(F)(F)F | *Doebelin et al., 2016* DOI: 10.1002/ cmdc.201600491 | ID:SR18957 | |
| Chemical compound, drug | Fc1cccc(Cl)c1CCc2 ccc3ccn(c3c2)S(=O)(=O) c4cccc(c4)C(F)(F)F | *Doebelin et al., 2016* DOI: 10.1002/ cmdc.201600491 | ID:SR19081 | |
| Chemical compound, drug | Fc1cccc(Cl)c1COc2 ccc3CCN(c3c2)S(=O)(=O) c4cccc(c4)C(F)(F)F | *Doebelin et al., 2016* DOI: 10.1002/ cmdc.201600491 | ID:SR19170 | |
| Chemical compound, drug | FC(F)(F)c1cccc(c1) S(=O)(=O)N2CCc3ccc (OCc4c(Cl)cccc4Cl)cc23 | *Doebelin et al., 2016* DOI: 10.1002/ cmdc.201600491 | ID:SR19171 | |
| Chemical compound, drug | Cc1cccc(Br)c1COc 2ccc3CCN(c3c2)S(=O) (=O)c4cccc(c4)C(F)(F)F | *Doebelin et al., 2016* DOI: 10.1002/ cmdc.201600491 | ID:SR19172 | |
| Chemical compound, drug | Cc1cccc(C)c1COc 2ccc3CCN(c3c2)S (=O)(=O)c4cccc(c4) C(F)(F)F | *Doebelin et al., 2016* DOI: 10.1002/ cmdc.201600491 | ID:SR19173 | |

*Continued on next page*

*Continued*

| Reagent type (species) or resource | Designation | Source or reference | Identifiers | Additional information |
|---|---|---|---|---|
| Chemical compound, drug | COc1ccccc1COc2cc3CCN(c3c2)S(=O)(=O)c4cccc(c4)C(F)(F)F | *Doebelin et al., 2016* DOI: 10.1002/cmdc.201600491 | ID:SR19174 | |
| Chemical compound, drug | Fc1cccc(Cl)c1COc2ccc3ccn(c3c2)S(=O)(=O)c4cccc(c4)C(F)(F)F | *Doebelin et al., 2016* DOI: 10.1002/cmdc.201600491 | ID:SR19257 | |
| Chemical compound, drug | Cc1cccc(c1)S(=O)(=O)N2CCc3ccc(OCc4c(F)cccc4Cl)cc23 | *Doebelin et al., 2016* DOI: 10.1002/cmdc.201600491 | ID:SR19258 | |
| Chemical compound, drug | COc1cccc(c1)S(=O)(=O)N2CCc3ccc(OCc4c(F)cccc4Cl)cc23 | *Doebelin et al., 2016* DOI: 10.1002/cmdc.201600491 | ID:SR19263 | |
| Chemical compound, drug | Fc1cccc(Cl)c1COc2ccc3CCN(c3c2)S(=O)(=O)c4cc(Cl)cc(Cl)c4 | *Doebelin et al., 2016* DOI: 10.1002/cmdc.201600491 | ID:SR19264 | |
| Chemical compound, drug | FC(F)(F)c1cccc(c1)S(=O)(=O)N2CCc3ccc(OCc4c(Cl)cncc4Cl)cc23 | *Doebelin et al., 2016* DOI: 10.1002/cmdc.201600491 | ID:SR19265 | |
| Chemical compound, drug | COc1cccnc1COc2cc3CCN(c3c2)S(=O)(=O)c4cccc(c4)C(F)(F)F | *Doebelin et al., 2016* DOI: 10.1002/cmdc.201600491 | ID:SR19266 | |
| Chemical compound, drug | Fc1cccc(Cl)c1COc2ccc3CCN(c3c2)S(=O)(=O)c4cncc(c4)C(F)(F)F | *Doebelin et al., 2016* DOI: 10.1002/cmdc.201600491 | ID:SR19267 | |
| Chemical compound, drug | FC(F)(F)Oc1ccccc1COc2ccc3CCN(c3c2)S(=O)(=O)c4cccc(c4)C(F)(F)F | *Doebelin et al., 2016* DOI: 10.1002/cmdc.201600491 | ID:SR19268 | |
| Chemical compound, drug | CC(Oc1ccc2CCN(c2c1)S(=O)(=O)c3cccc(c3)C(F)(F)F)c4c(F)cccc4Cl | *Doebelin et al., 2016* DOI: 10.1002/cmdc.201600491 | ID:SR19269 | |
| Chemical compound, drug | Fc1cccc(Cl)c1COc2ccc3CCN(c3c2)S(=O)(=O)c4cccc(n4)C(F)(F)F | *Doebelin et al., 2016* DOI: 10.1002/cmdc.201600491 | ID:SR19344 | |
| Chemical compound, drug | Fc1cccc(Cl)c1COc2cc3CCN(c3c2)S(=O)(=O)c4cccc(Cl)c4 | *Doebelin et al., 2016* DOI: 10.1002/cmdc.201600491 | ID:SR19346 | |
| Chemical compound, drug | Fc1cccc(Cl)c1COc2ccc3CCN(c3c2)S(=O)(=O)c4cc(ccn4)C(F)(F)F | *Doebelin et al., 2016* DOI: 10.1002/cmdc.201600491 | ID:SR19348 | |
| Chemical compound, drug | Fc1cccc(Cl)c1COc2cc3CCN(c3c2)S(=O)(=O)c4ccnc(c4)C(F)(F)F | *Doebelin et al., 2016* DOI: 10.1002/cmdc.201600491 | ID:SR19349 | |
| Chemical compound, drug | COc1cc(Cl)ccc1COc2ccc3CCN(c3c2)S(=O)(=O)c4cccc(c4)C(F)(F)F | *Doebelin et al., 2016* DOI: 10.1002/cmdc.201600491 | ID:SR19350 | |
| Chemical compound, drug | COc1c(Cl)cccc1COc2ccc3CCN(c3c2)S(=O)(=O)c4cccc(c4)C(F)(F)F | *Doebelin et al., 2016* DOI: 10.1002/cmdc.201600491 | ID:SR19351 | |
| Chemical compound, drug | FC(F)(F)c1cccc(c1)S(=O)(=O)N2CCc3ccc(OCc4c(Cl)ccnc4Cl)cc23 | *Doebelin et al., 2016* DOI: 10.1002/cmdc.201600491 | ID:SR19352 | |
| Chemical compound, drug | COc1ccc(Cl)cc1COc2ccc3CCN(c3c2)S(=O)(=O)c4cccc(c4)C(F)(F)F | *Doebelin et al., 2016* DOI: 10.1002/cmdc.201600491 | ID:SR19353 | |

*Continued on next page*

*Continued*

| Reagent type (species) or resource | Designation | Source or reference | Identifiers | Additional information |
|---|---|---|---|---|
| Chemical compound, drug | Fc1cccc(Cl)c1C(=O)Nc2ccc3CCN(c3c2)S(=O)(=O)c4cccc(c4)C(F)(F)F | *Doebelin et al., 2016* DOI: 10.1002/cmdc.201600491 | ID:SR19355 | |
| Chemical compound, drug | CN(C(=O)c1c(F)cccc1Cl)c2ccc3CCN(c3c2)S(=O)(=O)c4cccc(c4)C(F)(F)F | *Doebelin et al., 2016* DOI: 10.1002/cmdc.201600491 | ID:SR19379 | |
| Chemical compound, drug | Fc1cccc(Cl)c1CNc2ccc3CCN(c3c2)S(=O)(=O)c4cccc(c4)C(F)(F)F | *Doebelin et al., 2016* DOI: 10.1002/cmdc.201600491 | ID:SR19381 | |
| Chemical compound, drug | CN(Cc1c(F)cccc1Cl)c2ccc3CCN(c3c2)S(=O)(=O)c4cccc(c4)C(F)(F)F | *Doebelin et al., 2016* DOI: 10.1002/cmdc.201600491 | ID:SR19382 | |
| Chemical compound, drug | CC1Cc2ccc(OCc3c(F)cccc3Cl)cc2N1S(=O)(=O)c4cccc(c4)C(F)(F)F | *Doebelin et al., 2016* DOI: 10.1002/cmdc.201600491 | ID:SR19425 | |
| Chemical compound, drug | Fc1cncc(Cl)c1COc2ccc3CCN(c3c2)S(=O)(=O)c4cccc(c4)C(F)(F)F | *Doebelin et al., 2016* DOI: 10.1002/cmdc.201600491 | ID:SR19497 | |
| Chemical compound, drug | FC(F)(F)c1cccc(c1)S(=O)(=O)N2CCc3ccc(OCc4c(Cl)cncc4C(F)(F)F)cc23 | *Doebelin et al., 2016* DOI: 10.1002/cmdc.201600491 | ID:SR19498 | |
| Chemical compound, drug | FC(F)(F)C1Cc2ccc(OCc3c(Cl)cncc3Cl)cc2N1S(=O)(=O)c4cccc(c4)C(F)(F)F | *Doebelin et al., 2016* DOI: 10.1002/cmdc.201600491 | ID:SR19546 | |
| Chemical compound, drug | Fc1cccc(Cl)c1COc2ccc3CC(N(c3c2)S(=O)(=O)c4cccc(c4)C(F)(F)F)C(F)(F)F | *Doebelin et al., 2016* DOI: 10.1002/cmdc.201600491 | ID:SR19547 | |
| Chemical compound, drug | Fc1cncc(Cl)c1COc2ccc3CC(N(c3c2)S(=O)(=O)c4cccc(c4)C(F)(F)C(F)(F)F | *Doebelin et al., 2016* DOI: 10.1002/cmdc.201600491 | ID:SR19548 | |
| Software, algorithm | HDX-Workbench | *Pascal et al., 2012* DOI: 10.1007/s13361-012-0419-6 | N/A | |
| Software, algorithm | Prism 5 | GraphPad | N/A | |
| Software, algorithm | R Studio | R studio Team (2015) | N/A | |
| Software, algorithm | PHENIX | *Adams et al., 2010* DOI: 10.1107/S0907444909052925 | N/A | |
| Software, algorithm | iMosflm | *Battye et al., 2011* DOI: 10.1107/s0907444910048675 | N/A | |

## Chemicals, Cloning, and Mutagenesis

Unless otherwise specified, all chemicals and reagents were purchased from Sigma-Aldrich (St. Louis, MO). Complementary DNA coding for residues 265–518 from human RORγ Variant 1 (Uniprot ID P51449) were cloned into the pESUMO-Pro (Lifesensor) vector using BsaI and XhoI (New England Biolabs). Mutant constructs were generated using a site-directed mutagenesis kit (Q5, New England Biolabs) using primers described in the supplement. Our RORγLBD-SRC2 cDNA was designed based on a previously published construct (*Li et al., 2017*), synthesized by Genscript, and cloned into the

pESUMO vector using the same enzymes as before. RORγHinge-LBD was cloned into the pBIND vector as previously described (*Kumar et al., 2010*). All plasmids were sequence verified.

## Protein expression and purification

All protein constructs were expressed in the *E. coli* strain BL21 Codon plus (DE3) RIL (invitrogen) cell line. The wild type and variant protein constructs were expressed by culturing *E. coli* in terrific broth supplemented with Carbenicillin (50 μg/L) in a temperature controlled orbital shaker (Innova) operating at 200 RPM at 37°C. After the culture reach an optical density ($OD_{610}$) of 0.5, the temperature of the incubation chamber was dropped to 16°C and IPTG was added to 250 μM. The cultures were then incubated for 16 hr until being harvested by centrifugation at 4°C. Cell pellets were then resuspended in ice cold phosphate buffered saline containing protease inhibitors (EDTA-free SigmaFast, Sigma-Aldrich) prior to subsequent harvesting, flash freezing in liquid nitrogen and storage at −80°C. Unless otherwise mentioned, all protein purification steps were conducted at 4°C. Cell pellets were resuspended with Ni-NTA buffer A (50 mM Tris pH 8.0, 500 mM NaCl, 10% (v/v) glycerol, and 25 mM imidazole) supplemented with protease inhibitors, DNase, and lysozyme. Cells were lysed using a French press operating at 20,000 psi. The crude cell lysate was clarified by centrifugation (40,000 rcf for 30 min) and the supernatant was filtered at 0.2 μm prior to loading onto Ni-NTA resin pre-equilibrated with buffer A using an AKTA protein purification system. Ni-NTA resin was washed with 10 column volumes of buffer A and protein was eluted using a continuous imidazole gradient with buffer B (50 mM Tris pH 8.0, 500 mM NaCl, 10% (v/v) glycerol, and 250 mM imidazole). The Ni-NTA eluate was concentrated using Amicon ultra 10 kDa molecular weight cutoff centrifugation tubes and injected onto an Superdex S200 (26/60) pre-equilibrated with buffer C (25 mM Hepes pH 7.4, 150 mM NaCl, 5% (v/v) glycerol, and 2 mM tris(2-carboxyethyl)phosphine (TCEP). Fractions containing the protein construct of interest were determined using sodium dodecyl phosphate-polyacrylamide gel electrophoresis (SDS-PAGE) and coomassie staining. The RORγLBD(265-507) and the RORγLBD-SRC2 construct was purified using a slightly different protocol. After the first Ni-NTA affinity step, the eluate protein was exchanged back into Ni-NTA buffer A using centrifugation tubes. The His-SUMO solubility tag was cleaved using His-tagged SUMO protease (Gift from *Mossessova and Lima, 2000*) by incubating at 4°C overnight while gently rocking in protein LoBind tubes (Eppendorf). The flow through of the second affinity step was collected and concentrated prior to loading onto a Superdex S200 (26/60) pre-equilibrated with buffer C. The gel filtration column eluate was then concentrated to 9 mg/mL and used immediately to set up crystallization trays. All protein products were verified using in-gel trypsin digestion and LC-MS/MS.

## X-ray crystallography

The RORγLBD-SRC2 construct was crystallized by sitting drops of 1 μL protein solution and 1 μL of reservoir solution containing 0.1 M sodium phosphate (pH 7.4), 0.5–0.7 M NaCl, 4–6% PEG4000, and 3% DMSO. Apo RORγLBD crystals formed after 3 day incubation at 22 °C. After growing for 2 weeks, crystals were picked and transferred for soaking in reservoir solution saturated with SR19265 or SR19355. Crystals were soaked overnight before being harvested, soaked in a cryoprotectant, and then stored in liquid nitrogen. Diffractions were collected at the advanced light source beam line 5.0.2. The RORγ(265-507) construct was purified and concentrated to ~12 mg/mL. Prior to setting up crystallization trays, SR2211 was added to three molar excess and the solution was incubated at 22°C for 1 hr. crystallization trays were prepared using 100–250 mM ammonium sulfate (Hampton Research), 10–25% PEG3350 (Hampton Research), and 100 mM Tris pH 8.0. Sitting drops were prepared using microbridges (Hampton research) by mixing 1.5 μL protein solution with 1.5 μL well solution and 0.1 μL of seed solution. Crystals grew over a 2 week period before harvesting, cryoprotection, and data collection at various synchrotron sources (*Supplementary file 1*). Datasets were processed using autoPROC using XDS as the data-processing engine (*Kabsch, 2010*). Initial phases of the structures were solved by the molecular replacement method using Phaser in Phenix 1.14 with PDB ID 5VB0 or 4MQ0 as the search model. Crystallographic refinement was performed using Phenix 1.14. Multiple cycle of manual rebuilding and structure model adjustment were carried out using the graphics program Coot (*Emsley and Cowtan, 2004*). Molecular figures were created using UCSF Chimera (*Pettersen et al., 2004*). Structure validation was carried out with MolProbity (*Chen et al., 2010*). Data processing and refinement statistics are shown in *Supplementary file 1*.

### HDX-MS

Differential HDX-MS experiments were conducted as previously described with a few modifications (*Chalmers et al., 2006*). Differential HDX-MS experiments that reported here are summarized in *Supplementary file 2*.

## Peptide Identification

Protein samples were injected for inline pepsin digestion and the resulting peptides were identified using tandem MS (MS/MS) with an Orbitrap mass spectrometer (Fusion Lumos, ThermoFisher). Following digestion, peptides were desalted on a C8 trap column and separated on a 1 hr linear gradient of 5–40% B (A is 0.3% formic acid and B is 0.3% formic acid 95% $CH_3CN$). Product ion spectra were acquired in data-dependent mode with a one second duty cycle such that the most abundant ions selected for the product ion analysis by higher-energy collisional dissociation between survey scan events occurring once per second. Following MS2 acquisition, the precursor ion was excluded for 16 s. The resulting MS/MS data files were submitted to Mascot (Matrix Science) for peptide identification. Peptides included in the HDX analysis peptide set had a MASCOT score greater than 20 and the MS/MS spectra were verified by manual inspection. The MASCOT search was repeated against a decoy (reverse) sequence and ambiguous identifications were ruled out and not included in the HDX peptide set.

## HDX-MS analysis

For differential HDX, RORγLBD - ligand complexes was formed by incubating RORγLBD (5 µM) with compounds (50 µM) for 1 hr on ice. Next, 5 µLof sample was diluted into 20 µL $D_2O$ buffer (50 mM sodium phosphate pH 7.4, 150 mM NaCl, 2 mM TCEP) and incubated for various time points (0, 10, 30, 110, 380, 1270, 4270, and 14400 s) at 4°C. The deuterium exchange was then slowed by mixing with 25 µL of cold (4°C) 3M urea and 1% trifluoroacetic acid. Quenched samples were immediately injected into the HDX platform. Upon injection, samples were passed through an immobilized pepsin column (2 mm ×2 cm) at 50 µL min$^{-1}$ and the digested peptides were captured on a 2 mm ×1cm C8 trap column (Agilent) and desalted. Peptides were separated across a 2.1 mm ×5cm C18 column (1.9 µL Hypersil Gold, ThermoFisher) with a linear gradient of 4–40% $CH_3CN$ and 0.3% formic acid, over 5 min. Sample handling, protein digestion and peptide separation were conducted at 4°C. Mass spectrometric data were acquired using an Orbitrap mass spectrometer (Fusion Lumos, ThermoFisher). The intensity weighted mean m/z centroid value of each peptide envelope was calculated and subsequently converted into a percentage of deuterium incorporation. This is accomplished determining the observed averages of the undeuterated and fully deuterated spectra and using the conventional formula described elsewhere (*Zhang and Smith, 1993*). Corrections for back-exchange were determined empirically using a $D_{max}$ control. Briefly, this sample was generated by mixing 5 µL of protein sample with 20 µL of deuterated buffer and incubated at 37°C overnight before being queued to for subsequent quenching, and injection.

## Data rendering

The HDX data from all overlapping peptides were consolidated to individual amino acid values using a residue averaging approach. Briefly, for each residue, the deuterium incorporation values and peptide lengths from all overlapping peptides were assembled. A weighting function was applied in which shorter peptides were weighted more heavily and longer peptides were weighted less. Each of the weighted deuterium incorporation values were then averaged to produce a single value for each amino acid. The initial two residues of each peptide, as well as prolines, were omitted from the calculations. This approach is similar to that previously described (*Keppel and Weis, 2015*). HDX analyses were performed in triplicate, with single preparations of each purified protein/complex. Statistical significance for the differential HDX data is determined by t-test for each time point and is integrated into the HDX Workbench software (*Pascal et al., 2012*).

### Thermal shift assay

Differential Scanning Fluorimetry based thermal shift assays were conducted using GloMelt reagents (Biotium) according to manufacturer's protocol with slight modifications. Briefly, 10 µL of 4 µM RORγ LBD in assay buffer (25 mM HEPES pH 7.4, 150 mM NaCl, 5 mM DTT) were added to a 384 well

plate (Applied Biosystems). 100 nL of 200X compound or vehicle control solutions were added using a BioMeck NX$^P$ pintool to a final concentration of 10 µM. The protein-compound solutions were mixed with 10 µL of 2X glomelt solution containing 2 µM ROX reference dye and incubated for 30 min at room temperature. Plates were read on a 7900HT Fast Real-Time PCR System (Applied Biosystems) and data were analyzed using GraphPad Prism 5.

## Coactivator recruitment assays

AlphaScreen-based peptide recruitments assays were performed using streptavidin coated donor beads and anti-His-tag antibody coated AlphaLisa acceptor beads (Perkin Elmer). The assays were very sensitive to the distance of the N-terminus to the LXXLL NR-box motif and the use of a flexible Ahx linker. Briefly, 10 µL of a 100 nM HisSUMO-RORγLBD in assay buffer (25 mM HEPES pH 7.4, 150 mM NaCl, 5 mM DTT, 0.01% NP40) were plated in a black 384 well low-volume plate (Greiner Bio-One). 100 nL of 200X compound or vehicle control solutions were added using a Bio-Meck NX$^P$ pintool, 5 µL of a 200 nM Biotin-Ahx-SRC1-2 was added and the 15 µL solution was incubated at room temperature for 30 min, after which, 5 µL of 4X bead cocktail was added. The bead slurry was incubated at RT for 30 min before reading on an EnVision plate reader (Perkin Elmer).

Fluorescence polarization-based peptide recruitment assays were performed in 20 µL reaction volumes in low volume 384 well plates offered by Greiner. Fluorescent peptides were labeled at the N-terminus with FITC (Lifetein). Briefly, 10 µL of 2x protein (180–0 µM) solution in assay buffer (25 mM HEPES pH 7.4, 150 mM NaCl, 5 mM DTT, 0.01% NP40) were added to a black 384 well plate (Greiner). 4x compound solutions were made by diluting a 10 mM stock in DMSO into assay buffer and added to a final concentration of 100 µM. After a 15 min incubation period at room temperature, FITC-SRC1-2 was added to 100 nM in a 5 µL 4x solution in assay buffer. Fluorescence polarization was read on a Synergy NEO plate reader (BioTek) and the results were analyzed using GraphPad Prism 5 software.

## Cell-based promoter-reporter assays

HEK293T cells were obtained from a female fetus (ATCC, Cat:CRL-3216), and were validated using STR profiling. Mycoplasm tests were performed every 6 months. The pBIND-RORγHinge-LBD plasmid was cotransfected with UAS-LUC and TK-LUC plasmids into HEK293T cells using ExtremeGene 9 transfection regeant (Roche). After overnight incubation, cells were seeded into 96 well plates and compounds or vehicle controls were added 6 hr later. Dual luciferase assays were performed using Dual-Glo firefly and renilla luciferase reagents according to the manufacturer's protocol. Results were analyzed using GraphPad Prism software. Twenty-four hours after treatment, the luciferase activity was measured using the BriteLite or Dual-Glo luciferase assay system (Promega). Results were analyzed using GraphPad Prism 5 software.

## Statistical treatments and analysis

No statistical analysis was done to estimate sample sizes. For biochemical and cell-based assays, we define technical replicates as assay replicates and biological replicates as separate preparations of the protein or cells. Unless specified otherwise, all biochemical assays were performed with triplicate technical replicates and at least two biological replicates. Fluorescence polarization assays were performed with two technical replicates per condition. While full time course HDX-MS was conducted with three technical replicates, two time point HDX-MS screening of RORγ modulators was done with six technical replicates per compound and repeated on a separate batch of protein. Cell-based activity assays were performed with six technical replicates and two biological replicates. To perform the correlation and covariation analysis, R version 3.5.0 was used to analyze a collection of HDX-Workbench outputs from 38 modulator differential experiments. The results of this analysis are shown in *Supplementary file 3* and *Figure 4—figure supplement 1*. After extracting perturbation values (Δ%D) for 54 peptides at two timepoints for 38 compounds, a two tailed t test was applied to determine likelihood that the values were significantly different from 0. After applying a Benjamini-Hochberg multiple test correction (*Benjamini and Yekutieli, 2001*), only peptide-timepoints combinations where 15 or more compounds had perturbation values significantly different from 0 (false discovery rate <0.05) were kept. Using this reduced dataset, a Pearson correlation analysis comparing perturbation values to functional assay activity values was conducted using the Psych package.

Again, the multiple test correction method of Benjamini-Hochberg was applied and the resulting adjusted p values and R (*Takeda et al., 2012*) were visualized in excel. Selected correlations were manually charted in GraphPad Prism 5. The PerformanceAnalytics package was used to generate plots to visualize covariation within the reduced HDX-MS data set. The correlogram was generated using the corrplot package in a way that only showed R (*Takeda et al., 2012*) values whose corresponding covariation adjusted p values was less than 0.01. Selected covariation plots were manually generated in GraphPad Prism 5.

---

## Additional information

### Funding

| Funder | Author |
| --- | --- |
| National Cancer Institute | Patrick R Griffin |
| National Institute of General Medical Sciences | Timothy S Strutzenberg |

The funders had no role in study design, data collection and interpretation, or the decision to submit the work for publication.

### Author contributions

Timothy S Strutzenberg, Conceptualization, Data curation, Formal analysis, Investigation, Writing—original draft, Writing—review and editing; Ruben D Garcia-Ordonez, Data curation, Formal analysis, Investigation, Methodology; Scott J Novick, Investigation, Methodology; HaJeung Park, Data curation, Formal analysis, Methodology; Mi Ra Chang, Conceptualization, Funding acquisition, Methodology; Christelle Doebellin, Yuanjun He, Rémi Patouret, Resources, Investigation, Methodology; Theodore M Kamenecka, Resources, Formal analysis, Supervision, Funding acquisition, Methodology; Patrick R Griffin, Conceptualization, Formal analysis, Supervision, Funding acquisition, Writing—original draft, Project administration, Writing—review and editing

### Author ORCIDs

Timothy S Strutzenberg [iD] https://orcid.org/0000-0003-0598-534X
Patrick R Griffin [iD] https://orcid.org/0000-0002-3404-690X

### Decision letter and Author response

Decision letter https://doi.org/10.7554/eLife.47172.019
Author response https://doi.org/10.7554/eLife.47172.020

---

## Additional files

### Supplementary files

• Supplementary file 1. X-ray Co-crystallographic statistics (related to *Figures 1* and *2*). Data collection and refinement statistics are shown for SR2211, SR19265, and SR19355 co-crystal structures.
DOI: https://doi.org/10.7554/eLife.47172.012

• Supplementary file 2. Summary of full time course differential HDX-MS (related to *Figures 1* and *2*). Data for control and test conditions were averaged across all timepoints and subtracted to generate differential values. Error was propagated and is shown in parentheses.
DOI: https://doi.org/10.7554/eLife.47172.013

• Supplementary file 3. The two timepoint HDX-MS screening data were treated and summarized in an. xlsx file (related to *Figures 3* and *4*). The DMSO tab shows values for apo protein collected for every compound. The Differential tab shows the Δ%D value collected for each compound. The Differential_SEM tab shows the propagated standard error of the mean (SEM) for each measurement. The ExperimentalData and ExperimentalData_SEM tabs show the biochemical data measurements and SEM for each compound that was used for correlation analysis. The ReducedDifferential and

ReducedDifferential_SEM tab shows perturbation values and SEM that are statistically significant from 0 with multiple testing correction FDR > 0.05 and absolute value greater than 5 Δ%D.
DOI: https://doi.org/10.7554/eLife.47172.014

• Transparent reporting form DOI: https://doi.org/10.7554/eLife.47172.015

### Data availability

Due to their large size, all the raw data from MS analysis is available from the authors. HDX Workbench outputs files have been uploaded to figshare (10.6084/m9.figshare.8230685) along with a treated data summary which summarizes the findings from the raw data.

The following dataset was generated:

| Author(s) | Year | Dataset title | Dataset URL | Database and Identifier |
|---|---|---|---|---|
| Timothy S Strutzenberg, Ruben D Garcia-Ordonez, Scott J Novick, HaJeung Park, Mi Ra Chang, Christelle Doebellin, Yuanjun He, Rémi Patouret, Theodore M Kamenecka, Patrick R Griffin | 2019 | HDX Workbench Export Data | https://doi.org/10.6084/m9.figshare.8230685 | figshare, 10.6084/m9.figshare.8230685 |

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
