## [Decision Letter]

Thank you for submitting your article "HDX-MS reveals structural determinants for RORγ hyperactivation by synthetic agonists" for consideration by *eLife*. Your article has been reviewed by two peer reviewers, and the evaluation has been overseen by a Reviewing Editor and Philip Cole as the Senior Editor. The reviewers have opted to remain anonymous.

The reviewers have discussed the reviews with one another and the Reviewing Editor has drafted this decision to help you prepare a revised submission.

Summary:

This interesting paper uses a combination of four complementary approaches to explore the activation mechanism of the nuclear receptor RORγ: segmental dynamics assessed by HDX; mutagenesis, comparison of various synthetic ligands, and crystal structures of three ligands in complex with the receptor.

Major strengths of the paper are the importance of the question been asked and the translatability of the findings to other nuclear receptors where differential responses to ligands are observed but for which mechanism is lacking. The data presented are generally rigorous and convincing. What is especially interesting is the correlation between the HDX data of different regions of the LBD.

A limitation of the study is that "activation" is defined in the context of a synthetic reporter system in transfected cells. There are no experiments to address the biological significance of the relationship between structure and activity reported. However, the reviewers and editor agree that the study sufficiently advances the field to justify publication in the absence of major additional work. Textual revisions are requested as outlined below.

Essential revisions:

1) One of the important conclusions that the authors reach is that the relatively high "basal" activity observed in vivo for RORγ is due to constitutive binding of an endogenous ligand which they term a silent agonist. This argument is based on a mutation that reduces the intrinsic basal activity of the receptor, makes the receptor refractory to activation by endogenous ligands, but still enables the receptor to respond to certain synthetic ligands. For me this is a plausible explanation, but not proof. The mutation could be responsible for two independent consequences, i.e. loss of intrinsic activity and loss of endogenous ligand binding. It is not proven that these are linked. This caveat should be acknowledged and discussed.

2) The authors present the two-step activation resulting from ligand binding as if they are the first to propose this. The concept that the ligand first stabilizes the global fold of the LBD and then repositions H12 has been understood in the field for many years and exemplified by human mutations that affect both processes. This should be acknowledged and discussed.

---

## [Author Response]

Essential revisions:1) One of the important conclusions that the authors reach is that the relatively high "basal" activity observed in vivo for RORγ is due to constitutive binding of an endogenous ligand which they term a silent agonist. This argument is based on a mutation that reduces the intrinsic basal activity of the receptor, makes the receptor refractory to activation by endogenous ligands, but still enables the receptor to respond to certain synthetic ligands. For me this is a plausible explanation, but not proof. The mutation could be responsible for two independent consequences, i.e. loss of intrinsic activity and loss of endogenous ligand binding. It is not proven that these are linked. This caveat should be acknowledged and discussed.

We agree that the data presented in this manuscript cannot be taken as proof of endogenous ligand requirement by the WT protein. We have tried to address the concern that the mutation alters the intrinsic activity of the protein via functional assays and while these measurements support the proposed model, they fail to directly prove that the endogenous ligand is the driving observed basal activity. We have added these comments to the Discussion section in the second paragraph. Below is the text that discusses these concerns:

“To address the ligand-dependent activity of RORγ, we employed a ‘bump-and-hole’ strategy where A368V RORγLBD does not respond to endogenous ligand and has little to no intrinsic activity in promotor-reporter assays. […] Taken together, these observations are collectively consistent with the model where endogenous ligands are required for the observed activity in cells.”

2) The authors present the two-step activation resulting from ligand binding as if they are the first to propose this. The concept that the ligand first stabilizes the global fold of the LBD and then repositions H12 has been understood in the field for many years and exemplified by human mutations that affect both processes. This should be acknowledged and discussed.

We thank the reviewers for pointing out that we have presented the two-stage activation model so arrogantly. We have revised the text to highlight that this model is common and referred to text that discusses these models in more detail. Specifically, the text that we have added to the third paragraph of the Discussion section is below:

“While this model of NR activation is commonly observed throughout the NR structural biology field, we have elaborated on the model to elucidate the structural determinants for the gradient of observed activities exhibited by RORγ agonists.”